

# Association between yoga and related contextual factors with moderate-to-vigorous physical activity among children and youth aged 5 to 17 years across five Indian states

Jamin Patel[1,2], Sheriff Ibrahim[2], Jasmin Bhawra[3,4],
Anuradha Khadilkar[3] and Tarun Reddy Katapally[1,2,3,5]

[1] Department of Epidemiology and Biostatistics, Schulich School of Medicine and Dentistry, University of Western Ontario, London, Ontario, Canada
[2] Faculty of Health Sciences, Western University, DEPtH Lab, London, Ontario, Canada
[3] Hirabai Cowasji Jehangir Medical Research Institute, Pune, Maharashtra, India
[4] School of Occupational and Public Health, Toronto Metropolitan University, Toronto, Ontario, Canada
[5] Lawson Health Research Institute, Children's Health Research Institute, London, Ontario, Canada

Corresponding author
Tarun Reddy Katapally,
tarun.katapally@uwo.ca

## ABSTRACT

Physical inactivity is one of the four key preventable risk factors, along with unhealthy diet, tobacco use, and alcohol consumption, underlying most noncommunicable diseases. Promoting physical activity is particularly important among children and youth, whose active living behaviours often track into adulthood. Incorporating yoga, an ancient practice that originated in India, can be a culturally-appropriate strategy to promote physical activity in India. However, there is little evidence on whether yoga practice is associated with moderate-to-vigorous physical activity (MVPA) accumulation. Thus, this study aims to understand how yoga practice is associated with MVPA among children and youth in India. Data for this study were obtained during the coronavirus disease lockdown in 2021. Online surveys capturing MVPA, yoga practice, contextual factors, and sociodemographic characteristics, were completed by 5 to 17-year-old children and youth in partnership with 41 schools across 28 urban and rural locations in five states. Linear regression analyses were conducted to assess the association between yoga practice and MVPA. After controlling for age, gender, and location, yoga practice was significantly associated with MVPA among children and youth ($\beta = 0.634$, $p < 0.000$). These findings highlight the value of culturally-appropriate activities such as yoga, to promote physical activity among children and youth. Yoga practice might have a particularly positive impact on physical activity among children and youth across the world, owing to its growing global prevalence.

## INTRODUCTION

Physical activity is a key determinant in preventing and managing non-communicable diseases (NCDs) (*Lee et al., 2012*; *Biswas et al., 2022*; *Santos et al., 2023*; *Warburton, Nicol & Bredin, 2006*). Given that low-and middle-income countries bear a high proportion of the world's NCD burden, promoting physical activity among children and youth in the global south is crucial (*World Health Organization, 2009*, *2002*). With recent reports indicating that India is the most populous nation in the world (*Hertog & Gerland, 2023*; *Sciubba, 2023*), it is essential to address physical inactivity among Indian children and youth to minimize the global burden of NCDs. Studies have consistently proven that culturally-appropriate interventions can reduce health disparities and minimize the prevalence of NCDs among individuals living in India (*Joo & Liu, 2021*; *Natesan et al., 2015*; *Patel et al., 2017*).

Yoga originated in India as a practice for integrating the body and mind (*Basavaraddi, 2023*). It is a significant cultural component of physical activity that is widely promoted by various public and private organizations (*Bhawra et al., 2023*). Yoga practice includes participation in physical postures (asanas), breathing techniques (pranayama), and meditation practices (*Govindaraj et al., 2016*). Research has demonstrated the effectiveness of school-based yoga programs for Indian children and youth in improving mental health and cognitive performance (*Kauts & Sharma, 2009*; *Singh, 2018*; *Hart et al., 2022*; *Telles et al., 2014*). Yoga practice also enhances the musculoskeletal system, including improving muscle strength, manual dexterity, and grip strength (*Raghuraj & Telles, 1997*; *Mandanmohan et al., 2003*). Despite its numerous health benefits, yoga was one of India's lowest-ranked indicators of active living among children and youth, according to the 2022 India Report Card on physical activity for children and adolescents, as less than a quarter of them reported engaging in daily yoga practice (*Bhawra et al., 2023*). In contrast, the adoption of yoga has been increasing, with approximately 300 million people reporting practicing yoga worldwide (*Singh, 2022*).

Given the rising global prevalence of yoga, it is important to understand yoga's association with moderate-to-vigorous physical activity (MVPA), a key determinant in NCD prevention (*CDC, 2022*; *24-Hour Movement Guidelines, n.d*; *Participaction, 2022*). Current research on the effects of yoga on children and youth has been notably limited, especially considering that it is a culturally-appropriate intervention that could significantly benefit Indian children and youth (*Kauts & Sharma, 2009*; *Singh, 2018*; *Hart et al., 2022*; *Telles et al., 2014*). Thus, this study aims to investigate the association between yoga practice and MVPA among children and youth in India, while controlling for sociodemographic and related contextual factors, to inform culturally-appropriate physical activity interventions.

## MATERIALS AND METHODS

### Design

Data were collected in this study from multiple centres using digital surveys during the coronavirus disease lockdown in India in 2021 (*Bhawra et al., 2023*; *Vispute et al., 2023*).

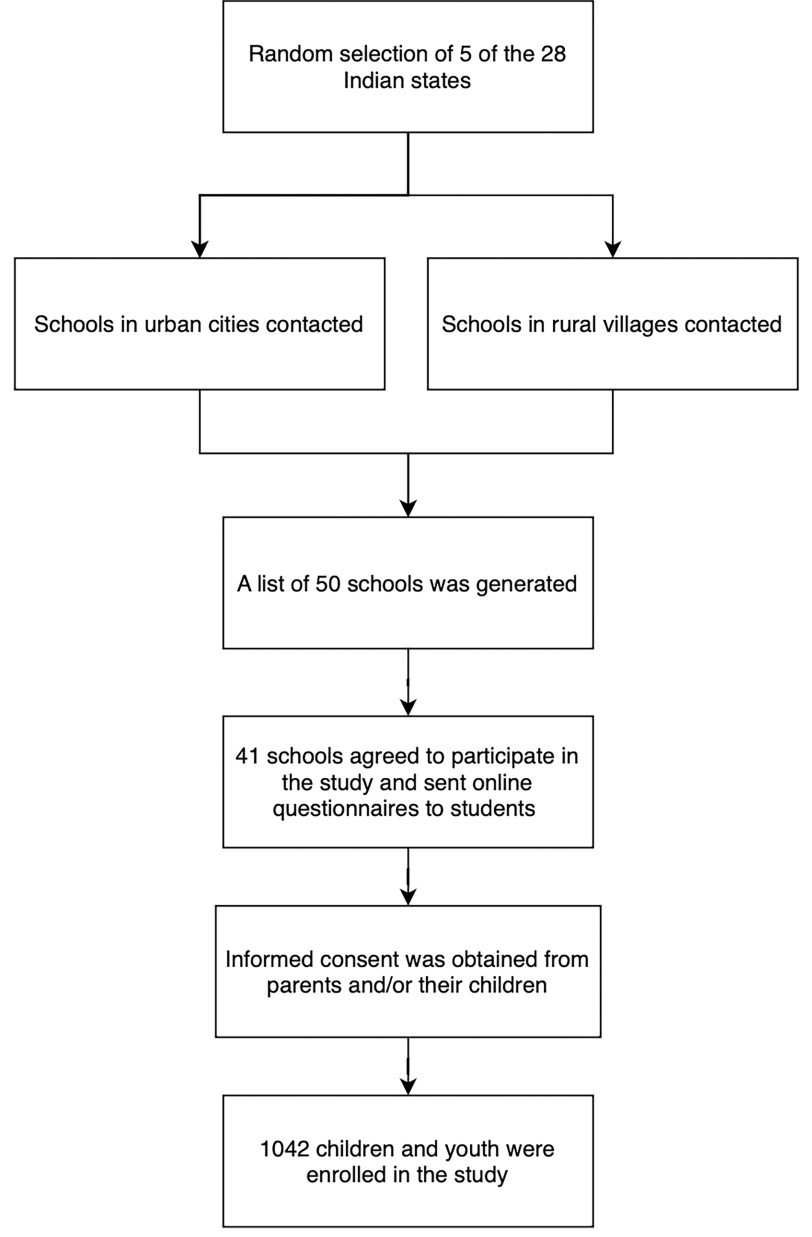

**Figure 1 Diagram of multi-stage random sampling method.**

The study employed a cross-sectional, observational design, and utilized a multi-stage random sampling method (Fig. 1) to select schools and recruit children and youth. Ethics approval was obtained from the Ethics Committee of Jehangir Clinical Development Centre Pvt. Ltd in Pune, Maharashtra (EC registration number—ECR/352/Inst/MH/2013/ RR-19).

## Recruitment and participants

The multi-staged stratified random sampling procedure involved randomly selecting five out of the 28 Indian states (Gujarat, Madhya Pradesh, Maharashtra, Tamil Nadu, and

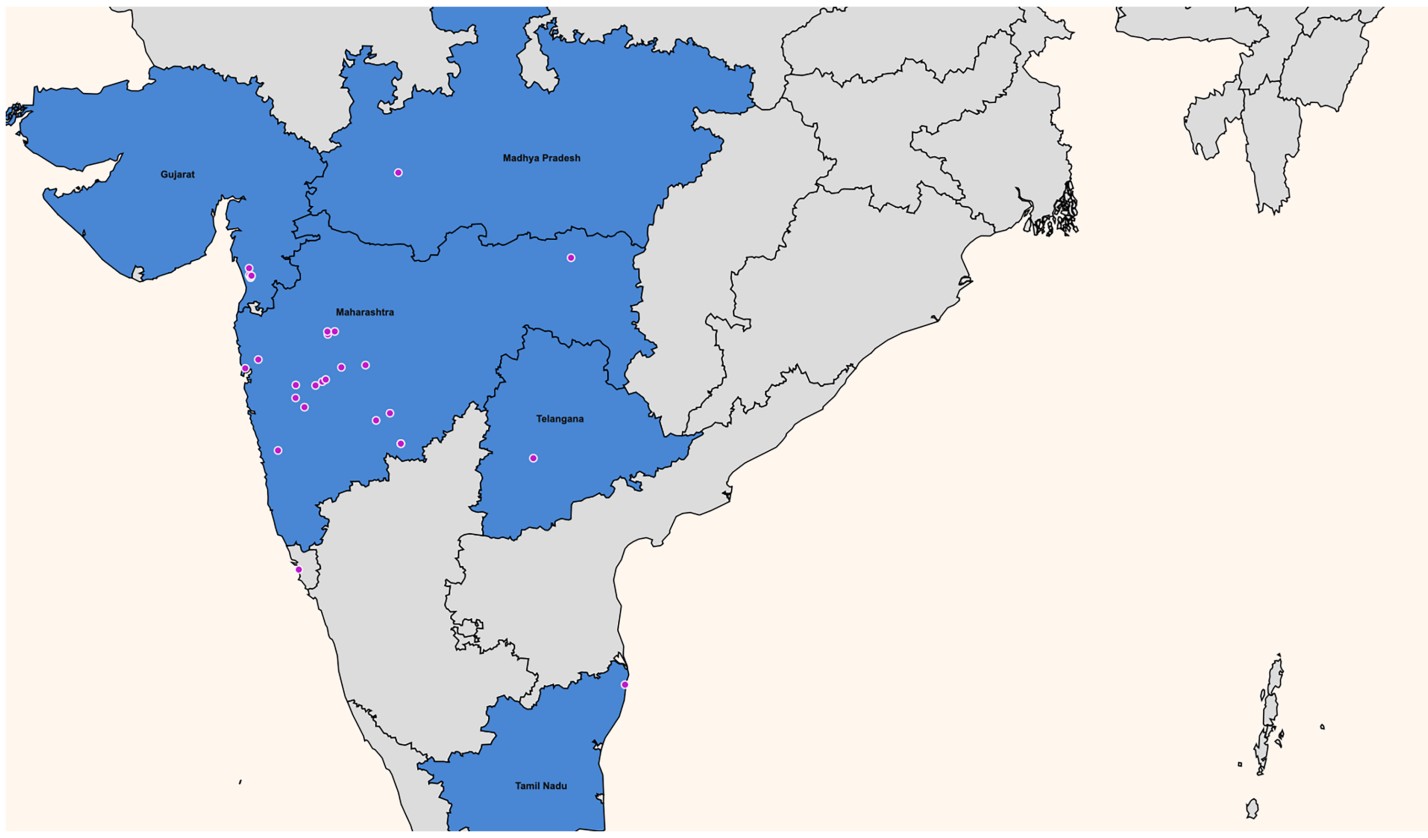

**Figure 2 Map of the 28 cities and villages across five Indian states where children and youth were recruited.** Map created in MapGeo—Interactive Geo Maps.

Telangana). Then, a city (urban area) and a neighbouring village (rural area) were randomly selected from each state (Fig. 2). A list of 50 schools was generated, with village schools being government public schools. School principals were approached with study information and invited to participate in the study. Of the 50 schools invited, 41 schools participated, with the primary reason for refusal being that the schools did not have time to participate in the study. Due to the ongoing coronavirus disease lockdown imposed in India in February 2021 (*Ghosal, 2021*), the principals electronically shared the study information and consent forms with students and their parents. The inclusion criteria for this study were apparently healthy students aged 5 to 17. Children and youth with any serious illnesses or chronic disorders were excluded from the study. Before emailing (between March 15, 2021, and May 20, 2021) digital survey links to students, online written informed consent was obtained from all students and their parents. Parents of children younger than 13 years were requested to assist their children in self-reporting information. A sample size calculation was carried out to achieve a 95% confidence level and a 5% margin of error. The sample size calculation resulted in a sample of 385 participants. Nevertheless, this study engaged with a total of 1,042 children. Data were collected to capture contextual factors within the following domains: physical activity and sedentary behaviour accumulation, yoga practice, participation in sports, active

transportation, peer support for physical activity, built environment, and school infrastructure.

## MEASURES

### Sociodemographic characteristics

As a part of this study, children and youth were asked to answer questions pertaining to their age, gender, and geographic location. The date of birth was collected in the questionnaire to identify the age of participants. We further categorized individuals into three age groups (aged 5 to 10 years, aged 11 to 13 years, and aged 14 to 17 years), representing the different developmental stages from childhood to adolescence. In particular, the World Health Organization defines individuals under 10 years of age as children, whereas individuals over 10 years of age are considered adolescents (*World Health Organization, n.d*). *Llorente-Cantarero et al. (2020)* found that physical activity and sedentary behaviour increased in the transition from childhood to adolescence. Moreover, there are distinct physical, sexual, cognitive, social, and emotional changes that occur between early adolescence (ages 10 to 13) and middle adolescence (ages 14 to 17) that may influence physical activity (*HealthyChildren.Org, 2019*). Gender was collected by asking participants whether they identified as "male" or "female". Geographic location was determined based on whether the school participants attended was classified as urban or rural based on its distance from the nearest population centre.

### Physical activity

Physical activity was assessed using a modified version of the International Physical Activity Questionnaire—Short Form (*Craig et al., 2003*), which asked participants about the weekly frequency and duration in minutes per day spent on various activities, including sports, weight training, running, and other disparate activities, such as jumping ropes, playing garden games, and swimming. The physical activity data were categorized into three intensity types: inactivity, light activity, and MVPA. Studies have found the Intraclass Correlation Coefficient for self-reported MVPA among children and youth to range from fair to good (range 0.565–0.78) (*Ng et al., 2019*; *Strugnell et al., 2014*). MVPA was calculated as the combined time spent in moderate and vigorous activities. The full physical activity questionnaire, including the list of activities, can be found in Table S1.

### Yoga

The study utilized a questionnaire to assess two aspects of yoga practice: "breathing yoga" and "physical yoga". Breathing yoga encompassed pranayama practice and deep breathing exercises (*Himalayan Yoga Institute, n.d*), while physical yoga included yoga stretches and various practices such as hatha, vinyasa, and ashtanga yoga (*Taylor, 2022*). Participants reported the daily duration and weekly frequency of engaging in both breathing yoga and physical yoga.

## Peer support

Peer support was assessed by asking participants about the number of close physically active friends they have, with response options ranging from zero to four or more. Responses were categorized as either "Has no active friends" if the response was zero or "Has one or more active friends" if the response was between one and four.

## School infrastructure

The perception of school infrastructure, including opportunities and resources for engaging in physical activity, was measured by asking participants whether or not their school organizes physical activity before school hours, during lunch hours, or after school hours, and whether their school has sports/physical activity competitions with other schools.

## Statistical analysis

All analyses were conducted in R 4.2.2 (*R Core Team, 2021*) 12.0 + 353. The independent and dependent variables were the average minutes of yoga practice and MVPA per day, respectively. The overall sample was further stratified into subgroups based on gender (male and female), geographic location (urban and rural), and age cohort (5 to 10 years old, 11 to 13 years old, and 14 to 17 years old), resulting in eight multiple linear regression models. Models were adjusted for age, school infrastructure (all models), peer support (all models), gender (overall and location models), and location (overall and gender models). Welch's unpaired sample t-tests for unequal variances, one-way analysis of variance (ANOVA), and *post-hoc* analyses using the Tukey-Kramer method for unequal variances were performed to compare the average yoga practice and MVPA across different factors, including gender, age, and location. All results mentioned in this study were considered statistically significant at a significance level of $p < 0.05$.

## RESULTS

Of the 1,042 children and youth that participated in this study, 992 participants were analyzed after excluding participants that were not aged 5 to 17. Table 1 presents the full sample summary, including sociodemographic and contextual factors. The study involved a diverse sample, consisting of males (50.3%) and females (49.7%), as well as a combination of urban (59.7%) and rural (40.3%) residents. The sample included 34.2% of participants aged 5 to 10, 32.1% aged 11 to 13, and 33.8% aged 14 to 17. Further, 92.9% of the participants had at least one active friend, 63.0% reported that their school organizes physical activity programs, and 86.6% reported that their school participates in sports competitions with other schools.

Table 2 shows the average yoga practice and MVPA levels of children and youth who participated in this study. The mean minutes spent daily on MVPA was 82.51 and the daily minutes of yoga practice was 8.61. Males had significantly higher MVPA than females (t = 10.57, df = 917.92, $p < 0.000$). In contrast, females spent more time on yoga practice than males (t = 2.20, df = 893.76, $p = 0.028$). Moreover, children and youth living in urban

**Table 1 Summary statistics of the children and youth who participated in the study.**

| Category | Sample size | Proportion (percentage) |
|---|---|---|
| **Age (N = 992)** | | |
| 5–10 | 339 | 34.2 |
| 11–13 | 318 | 32.1 |
| 14–17 | 335 | 33.8 |
| **Gender (N = 992)** | | |
| Male | 499 | 50.3 |
| Female | 493 | 49.7 |
| **Location (N = 992)** | | |
| Rural | 400 | 40.3 |
| Urban | 592 | 59.7 |
| **Active Friends (N = 990)** | | |
| No active friends | 70 | 7.1 |
| One or more active friends | 920 | 92.9 |
| **School organizes physical activity (N = 990)** | | |
| No | 366 | 37.0 |
| Yes | 624 | 63.0 |
| **School participates in sports competition with other schools (N = 883)** | | |
| No | 118 | 13.4 |
| Yes | 765 | 86.6 |
| **Overall** | 992 | N/A |

regions spent more time on yoga practice than their rural counterparts (t = 7.24, df = 950.29, $p < 0.000$). Additionally, there was a significant difference in yoga practice across the aged 5 to 10, 11 to 13, and 14 to 17 age groups (F = 4.23, df = 2.00, $p = 0.015$). *Post-hoc* analyses found that the aged 11 to 13 group practiced yoga more than the aged 5 to 10 group (mean difference = 4.18, $p = 0.032$).

The regression models conducted in this study are presented in Table 3. In the overall model, yoga practice was associated with higher MVPA levels (β = 0.634, 95% confidence interval (CI) [0.373–0.894], $p < 0.000$). Further, yoga practice was associated with MVPA in the aged 5 to 10 model (β = 0.785, CI= [0.263–1.308], $p = 0.004$) and the aged 14 to 17 model (β = 0.671, CI = [0.281–1.061], $p < 0.000$); however, this association was not significant in the aged 11 to 13 model. There was also a significant association between yoga practice and MVPA levels in the male (β = 0.757, CI = [0.246–1.268], $p = 0.004$), female (β = 0.578, CI = [0.303–0.853], $p < 0.000$), and urban (β = 0.674, CI = [0.372–0.975], $p < 0.000$) models, but not in the rural model.

Having active friends was associated with higher MVPA than having no active friends in the overall model (β = 39.145, CI = [19.126–59.165], $p < 0.000$). In the age models, having active friends was associated with higher MVPA in the aged 5 to 10 (β = 35.308, CI = [0.834–69.782], $p = 0.046$) and aged 11 to 13 (β = 46.889, CI = [12.236–81.542], $p = 0.008$)

**Table 2** Average daily minutes spent on yoga practice and MVPA of children and youth who participated in the study.

| Category | Mean minutes of MVPA daily (SD) | Mean minutes of yoga practice daily (SD) |
|---|---|---|
| **Age (N = 992)** | | |
| 5–10 | 76.777 (68.697) | 6.266 (16.044) |
| 11–13 | 87.732 (81.211) | 11.585 (25.555) |
| 14–17 | 83.429 (81.505) | 9.493 (22.120) |
| **Gender (N = 992)** | | |
| Male | 107.626 (80.680) | 7.259 (16.110) |
| Female | 57.797 (65.007) | 9.974 (22.191) |
| **Location (N = 992)** | | |
| Rural | 81.210 (73.978) | 3.813 (12.333) |
| Urban | 83.398 (79.546) | 11.857 (22.433) |
| **Active Friends (N = 990)** | | |
| No active friend | 37.033 (53.826) | 6.426 (15.975) |
| One or more active friends | 86.242 (77.719) | 8.794 (19.662) |
| **School organizes physical activity (N = 990)** | | |
| No | 70.034 (69.736) | 4.911 (15.967) |
| Yes | 90.186 (80.552) | 10.809 (20.903) |
| **School participates in sports competition with other schools (N = 883)** | | |
| No | 62.145 (69.225) | 6.683 (14.405) |
| Yes | 87.296 (79.492) | 8.962 (20.095) |
| **Overall** | 82.505 (77.292) | 8.608 (19.410) |

models; however, there was no significant association in the aged 14 to 17 model. A significant association was found between having active friends and MVPA levels in the male ($\beta$ = 42.600, CI = [2.485–82.715]), female ($\beta$ = 37.436, CI = [16.440–58.433], $p < 0.000$), and urban ($\beta$ = 42.565, CI = [20.641–64.489], $p < 0.000$) models.

Participants reporting that their school participated in sports competitions with other schools were associated with higher MVPA than those who did not in the overall ($\beta$ = 16.405, CI = [1.795–31.016], $p = 0.028$) and aged 5 to 10 ($\beta$ = 23.007, CI = [1.682–44.331], $p = 0.035$) models. Agreeing that their schools organized physical activity was associated with higher MVPA levels than participants who disagreed in the overall ($\beta$ = 17.477, CI = [7.005–27.949], $p = 0.001$) and aged 14 to 17 ($\beta$ = 22.805, CI = [6.242–39.368], $p = 0.007$) models. There was also a significant association between school-organized physical activity and MVPA in the male ($\beta$ = 16.804, CI = [0.740–32.868], $p = 0.041$), female ($\beta$ = 17.521, CI = [4.210–30.831], $p = 0.010$), urban ($\beta$ = 17.358, CI = [3.422–31.294], $p = 0.015$), and rural ($\beta$ = 19.706, CI = [3.901–35.512], $p = 0.015$) models.

**Table 3 Regression models showing relationship between yoga practice and moderate to vigorous physical activity.**

| | Overall[a] (Model 1) | 5–10 years[a] (Model 2) | 11–13 years[a] (Model 3) | 14–17 years[a] (Model 4) | Male[b] (Model 5) | Female[b] (Model 6) | Rural[c] (Model 7) | Urban[c] (Model 8) |
|---|---|---|---|---|---|---|---|---|
| Average minutes of yoga practice per day | 0.634* (0.373, 0.894) | 0.785* (0.263, 1.308) | 0.465 (−0.018, 0.948) | 0.671* (0.281, 1.061) | 0.757* (0.246, 1.268) | 0.578* (0.303, 0.853) | 0.341 (−0.258, 0.941) | 0.674* (0.372, 0.975) |
| Has no active friends – (Ref) | | | | | | | | |
| Has one or more active friends | 39.145* (19.126, 59.165) | 35.308* (0.834, 69.782) | 46.889* (12.236, 81.542) | 31.866 (−2.881, 66.613) | 42.600* (2.485, 82.715) | 37.436* (16.440, 58.433) | −3.555 (−79.557, 72.448) | 42.565* (20.641, 64.489) |
| School participates in sports competition with other schools—No (Ref) | | | | | | | | |
| Yes | 16.405* (1.795, 31.016) | 23.007* (1.682, 44.331) | 12.870 (−13.147, 38.887) | 6.686 (−24.401, 37.774) | 23.203 (−0.427, 46.832) | 8.325 (−9.246, 25.897) | 16.389 (−7.450, 40.228) | 16.150 (−2.825, 35.125) |
| School organizes physical activity—No (Ref) | | | | | | | | |
| Yes | 17.477* (7.005, 27.949) | 9.081 (−8.936, 27.097) | 18.995 (−1.298, 39.288) | 22.805* (6.242, 39.368) | 16.804* (0.740, 32.868) | 17.521* (4.210, 30.831) | 17.358* (3.422, 31.294) | 19.706* (3.901, 35.512) |
| Constant | −26.806 (−58.156, 4.544) | −35.280 (−94.406, 23.846) | −128.564 (−265.872, 8.744) | 76.768 (−53.906, 207.441) | 6.738 (−46.696, 60.173) | 14.787 (−22.076, 51.650) | 15.834 (−62.634, 94.302) | −22.568 (−60.809, 15.674) |
| N | 856 | 262 | 276 | 318 | 432 | 424 | 388 | 468 |

**Notes:**
Beta coefficients
(95% confidence interval) are reported.
* Indicates a statistically significant relationship at the $p < 0.05$ level.
[a] Models 1–4 were adjusted for age, gender, and location.
[b] Models 5 and 6 were adjusted for age and location.
[c] Model 7–8 were adjusted for age and gender.

## DISCUSSION

The relationship between physical inactivity and increased NCD risk among children and youth is well-documented (*Haileamlak, 2019*; *Biswas et al., 2022*; *Akseer et al., 2020*). This is an urgent concern in India where a significant number of children and youth do not meet the recommended physical activity guidelines (*Bhawra et al., 2023*). Given India's status as the most populous nation (*Hertog & Gerland, 2023*; *Sciubba, 2023*), addressing physical inactivity among Indian children and youth is crucial in minimizing the global NCD burden. As sociocultural factors play a significant role in determining the physical activity patterns of children and youth, it is imperative to adopt culturally-appropriate approaches to promote physical activity (*Bhawra et al., 2023*; *Hu et al., 2021*; *Bhawra et al., 2018*). Interventions that consider sociocultural factors can be more readily accepted, resulting in higher satisfaction in the target population (*Torres et al., 2020*). Being a valued cultural practice in India, yoga holds promise in reducing NCD risk and promoting healthy behaviours in children and youth (*Newcombe, 2017*). However, research on the effects of yoga on children and youth has been notably limited. By examining the effects of yoga practice on children and youth's MVPA levels, this study contributes to addressing this knowledge gap.

This study found that yoga practice was associated with higher MVPA levels among children and youth in diverse locations across India. Our study findings align with the 2022 India Report Card, which emphasizes promoting physical activity through yoga programs for children and youth. The report not only supports our main conclusion but also recommends strategies for increased engagement in yoga, highlighting the holistic benefits of integrating yoga into physical activity initiatives (*Bhawra et al., 2023*). Although no studies have examined the association between yoga practice and MVPA among children and youth in India, these findings are consistent with previous research that has demonstrated the positive effects of yoga interventions on various aspects of health among children and youth in India, including a study which found that yoga interventions can improve physical, cognitive, and emotional measures among children in India (*Telles et al., 2013*). In addition, a recent study by *Kasture et al. (2024)* found that yoga is comparable to physical exercise in influencing muscle function, highlighting the broader impact of yoga on health that goes beyond MVPA.

These findings can inform the Fit India Movement (*Ministry of Youth Affairs and Sports, 2020*), a national initiative focused on promoting physical activity and wellbeing, by emphasizing importance of culturally-appropriate activity such a yoga in improving MVPA.

This study also highlights nuanced variations in the relationship between yoga practice and MVPA. For instance, yoga practice was associated with MVPA in the 5 to 10-year age group and the 14 to 17-year age group, but not in the 11 to 13-year age group, indicating that incorporating yoga into physical activity interventions may have a differential impact on MVPA based on age. Moreover, this discrepancy may be attributable to the 11 to 13 age range representing a transition from childhood to adolescence, which involves significant

physical, social and cognitive changes. For instance, a study by *Cozett, Bassett & Leach (2016)* analyzed factors influencing physical activity among 11 to 13-year-olds and found that multiple factors are associated with changes in physical activity in this age group, such as parental influence, peer influence, perceived physical activity self-efficacy and perceived physical activity competence. Our study also found that the aged 11 to 13 group practiced yoga more than the aged 5 to 10 group, which could indicate a potential developmental trend in yoga practice as children grow older. These findings are significant, as previous studies have not compared yoga practice in children and youth across different age cohorts.

Further, having active friends was associated with higher levels of MVPA in different age groups. Having active friends was not only associated with higher MVPA in the overall sample but also among the 5 to 10-year and 11 to 13-year age groups, highlighting the importance of creating social support networks for physical activity among younger children. These findings are consistent with previous research highlighting peer support's importance in promoting physical activity among children and youth (*Bhawra et al., 2023*; *Leggett et al., 2012*; *Jago et al., 2011*). However, this association was not significant in the 14 to 17-year age group. This may be attributable to younger children engaging in more group-based physical activity, whereas older adolescents may have a higher degree of autonomy and independence in their activity choices, which may reduce the impact of peer influence. For instance, *Smith et al. (2022)* found that organized and team-based physical activity was more common among younger children, and the participation rates declined with chronological age. Interventions targeting social networks, including peer-led physical activity programs and initiatives encouraging joint physical activity with friends, could effectively address these gaps. It is also imperative to provide health education through individualized strategies for physical activity and to create opportunities during and after school for individual-based physical activities to promote MVPA among children and youth without active friends. One potential strategy to promote MVPA could be to encourage family support from siblings as it has been associated with higher physical activity levels among youth (*Bhawra et al., 2018*; *Sallis, Prochaska & Taylor, 2000*). Given the importance of social networks, future studies should examine how the number of siblings in the family is associated with MVPA levels, particularly among children and youth in India who do not have active friends. Moreover, current evidence suggests that social support from friends and family may enhance youth's experience with yoga and promote continued yoga practice (*Dai, Chen & Sharma, 2023*). Future studies should perform mediation analyses to examine the causal pathway between social support, yoga practice, and MVPA to add to our study findings on the associations between yoga and MVPA, and between active friends and MVPA.

The 2022 India Report Card also recommended incorporating yoga into physical education curriculums or after-school programs. This study highlights the association between school-supported extracurricular physical activity and MVPA levels, with children and youth attending a school participating in sports competitions with other schools or offering organized physical activity engaging in more MVPA than their counterparts who did not. Schools can play an important role in promoting physical

activity among children and youth by developing policies and programs, such as organized sports competitions between schools or regular physical activity breaks throughout the school day (*Bhawra et al., 2023*, *2018*). Moreover, variations in the association between school infrastructure and MVPA were found within the models segregated by age cohorts. For children aged 14 to 17, access to school-organized physical activity was associated with higher MVPA, emphasizing the value of implementing school-supported extracurricular physical activity such as structured physical activity programs for older youth. Similarly, access to school-based sports competitions was associated with higher MVPA in the 5 to 10 age group, indicating the effectiveness of providing such opportunities in promoting MVPA at a younger age. While the study's recommendations focus heavily on promoting physical activity among younger age groups, research has shown that physical activity habits formed at a young age can carry into adulthood, highlighting the importance of early intervention (*Ha et al., 2019*; *van Sluijs et al., 2021*; *Mathisen et al., 2023*).

This study is the first to examine yoga practice in children and youth residing in urban *versus* rural areas in India. Our results not only indicate that children and youth living in rural areas engage in less yoga than their urban counterparts, but also that yoga practice is only associated with higher MVPA in urban children and youth. These findings align with a previous study conducted by *Mishra et al. (2020)* which found that there was a higher prevalence of yoga practitioners in urban areas compared to rural areas of India. Given that yoga has ancient spiritual roots in India, where it was originally practiced in remote regions (*Burgin, 2007*), these findings highlight the contemporary shift in yoga's prevalence. This shift in yoga prevalence may be attributable to yoga being viewed as a lifestyle activity for physical and mental well-being, rather than solely limited to the spiritual and meditative importance (*Büssing et al., 2012*). Additionally, these disparities suggest unique challenges in promoting traditional modes of physical activity in rural areas. In urban areas, where organized sports and structured exercise programs are more accessible than in most rural areas, individuals practicing yoga are more likely to engage in other forms of physical activity (*Hadire & Pathak, 2020*). Promoting physical activity tailored to rural communities' unique needs and circumstances may be necessary to address these disparities. For instance, in rural areas, children and youth may be more involved in outdoor games and sports than their urban counterparts (*Kaur, 2023*). Community-based interventions that focus on creating safe spaces for physical activity in rural areas, such as playgrounds and sports fields could help to increase physical activity levels among children and youth living in rural areas.

The results also outlined gendered differences in yoga practice and MVPA levels. Male children and youth engaged in higher MVPA than females, while females engaged in more yoga than males. Several barriers may hinder male yoga practice, including the perception of yoga as a feminine and female-dominated activity (*Cagas, Biddle & Vergeer, 2021*). Previous studies found that more male children and youth in India met the MVPA guidelines than females (*Bhawra et al., 2023*; *Mathur et al., 2021*). However, this is the first study to identify gender differences in yoga practice among children and youth in India. Nevertheless, although engaging in yoga was associated with higher levels of MVPA for both males and females, the association was stronger for males. This may reflect differences
in male and female preferences and interest in distinct types of physical activity; however, further research is required to unpack these gendered differences. Based on our study findings, interventions tailored to gendered preferences, including culturally-appropriate yoga programs for children and youth, should be implemented in schools and community programs.

These findings are potentially generalizable to similar global populations practicing yoga. As an accessible physical activity option, yoga can be widely adopted by the youth population globally; however, future research is required to consider cultural and contextual factors. For instance, culturally-appropriate land-based active living initiatives that integrate culturally-appropriate physical activity have improved health outcomes among Indigenous Canadian youth (*Katapally, 2020*). Moreover, decolonized approaches that are well established in settler nations can inform culturally-relevant programing in previously colonized nations like India (*Ironside et al., 2020*; *Walker et al., 2021*; *Kannan et al., 2022*). Future research should examine the influence of other culturally-appropriate activities on MVPA levels, considering their potential to reduce health disparities and NCD prevalence in low- and middle-income countries such as India (*Joo & Liu, 2021*; *Natesan et al., 2015*; *Patel et al., 2017*).

## STRENGTHS AND LIMITATIONS

This study employed a robust stratified random sampling method across a large sample size to ensure a representative sample from both urban and rural areas, which enhances the generalizability of the current study findings. However, with India being a large country with a diverse population, it is important to note that the current study might not be representative of all Indian children and youth. The data collected during the Coronavirus disease pandemic offer unique insights into physical activity levels during a period of restrictions and limited outdoor opportunities; however, when interpreting the results, it is important to consider that they may differ in comparison to a non-pandemic time. Although self-reported measures indicate MVPA levels in children and youth in India, objective measures such as accelerometers and global positioning system monitors may reduce recall and social desirability biases as well as provide location-specific data (*Katapally, Bhawra & Patel, 2020*). Additionally, controlling for motivating and facilitating factors for physical activity is critical to ensure that the association between yoga practice and MVPA is valid. Future studies should substantiate our study findings with longitudinal studies and randomized controlled trials to control for additional unobserved confounders. Although previous research indicates there are varying levels of intensity to classify yoga (*i.e.*, asanas and pranayama) (*Sengupta, 2012*), these variations were beyond the scope of our study. Future studies should compare these associations between yoga practice and MVPA across varying intensities of yoga practice (*e.g.*, light activity for breathing yoga or moderate-to-vigorous activity for physical yoga) to capture nuanced variations in the relationship between yoga and MVPA.

This study did not distinguish between public and private educational institutions, which may be important for understanding the factors that are related to MVPA levels among youth. In particular, previous literature has found that in India, low socioeconomic

status has been associated with a decrease in physical activity (*Gulati et al., 2014*). Indian youth with higher family income were more likely to attend private school institutions, where students have been found to engage in less MVPA than their public school student counterparts (*Raskind et al., 2020*). Additionally, yoga practice could potentially be influenced by the type of school children and youth attend and the facilities provided. As this study did not capture whether yoga practice was accumulated as a part of a school physical activity program or whether their school participated in inter-school yoga competitions, biases related to the source and context of yoga engagement may have influenced the results. Future studies should examine how these associations between yoga practice and MVPA differ across various local school boards which may have varying physical activity promotion patterns.

## CONCLUSIONS

This study addresses a significant gap in the understanding of the relationship between yoga practice and engagement in MVPA. The evidence clearly indicates that yoga is a culturally-appropriate and effective means of promoting physical activity among children and youth in India. The results also support the need to develop age-, gender, and location-specific strategies, as well as the necessity to build social support and school infrastructure to promote MVPA. Nevertheless, with existing programs and policies struggling to achieve active living objectives, yoga can play an important role in addressing physical inactivity among children and youth in India-a critical factor in reducing the future burden of global NCDs.

## ACKNOWLEDGEMENTS

The authors acknowledge the research team at Hirabai Cowasji Jehangir Medical Research Institute, Pune, for obtaining these data during the coronavirus disease pandemic, as well as the families and schools that participated in this study.

### Funding

This study is supported by the Canada Research Chairs Program, which funds Dr. Tarun Katapally's research program. The funders had no role in study design, data collection and analysis, decision to publish, or preparation of the manuscript.

### Grant Disclosures

The following grant information was disclosed by the authors:
Canada Research Chairs Program.

### Competing Interests

The authors declare that they have no competing interests.

## Author Contributions

- Jamin Patel analyzed the data, prepared figures and/or tables, authored or reviewed drafts of the article, and approved the final draft.
- Sheriff Ibrahim analyzed the data, authored or reviewed drafts of the article, and approved the final draft.
- Jasmin Bhawra conceived and designed the experiments, performed the experiments, authored or reviewed drafts of the article, and approved the final draft.
- Anuradha Khadilkar conceived and designed the experiments, performed the experiments, authored or reviewed drafts of the article, and approved the final draft.
- Tarun Reddy Katapally conceived and designed the experiments, performed the experiments, authored or reviewed drafts of the article, and approved the final draft.

## Human Ethics

The following information was supplied relating to ethical approvals (*i.e.*, approving body and any reference numbers):

Ethics Committee of Jehangir Clinical Development Centre Pvt. Ltd in Pune, Maharashtra.

## Data Availability

The "Children and Youth in India 2021" dataset is available at Figshare: Patel, Jamin; Katapally, Tarun (2023). Children and Youth in India 2021: Yoga and MVPA. figshare. Dataset. https://doi.org/10.6084/m9.figshare.23586321.v2.

## Supplemental Information

Supplemental information for this article can be found online at http://dx.doi.org/10.7717/peerj.17369#supplemental-information.

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
