# Peer review of "Association between yoga and related contextual factors with moderate-to-vigorous physical activity among children and youth aged 5 to 17 years across five Indian states"

_PeerJ, doi:10.7717/peerj.17369_

## Round 0.1 · original submission · Major Revisions

Dear Dr. Patel and colleagues:

Thanks for submitting your manuscript to PeerJ. I have now received three independent reviews of your work, and as you will see, the reviewers raised some concerns about the research. Despite this, these reviewers are quite optimistic about your work and the potential impact it will have on research studying the health benefits and limitations of Yoga. Thus, I encourage you to revise your manuscript, accordingly, taking into account all of the concerns raised by the three reviewers.

Please ensure that your figures and tables contain all the information that is necessary to support your findings and observations.

Please edit the manuscript for clarity and typos. There appear to be some key references missing.

Please note that Reviewer 3 kindly provided a marked-up version of your manuscript.

Good luck with your revision,

-joe

Reviewer 1 ·

Basic reporting

No comments

Experimental design

1. The study should have been registered with CTRI (Clinical Trial Registry) India as all clinical trials including observational are mandated from 1st April 2018 to be registered with CTRI. Please refer for more details, https://ctri.nic.in/Clinicaltrials/faq.php#8a. If the study is exempted, please provide a reasonable explanation.
2. I recommend including ‘Inclusion and Exclusion criteria’ for the study group under “Material & Methods- Recruitment and participants” so this gives an idea of the type of schools data collected.

3. In Figure 2, study locations in the map, two circles close to northern regions of India (probably Delhi and Nepal) are also shown as study points. As per your manuscript, these places are not described anywhere. So, I suggest rechecking the figure and also describing what each color circle indicates.

4. In Table 1, the proportion of males and females needs to be rechecked.

Validity of the findings

No comments

Additional comments

This is a well-written and well-thought-out study.
The data and analysis are sound and the conclusions are valid.

Reviewer 2 ·

Basic reporting

Introduction:

a. Lane 33-34: Please explain this “However, there is little evidence of the relationship between yoga practice and physical activity”. According to us if we consider the definition of physical activity (PA) yoga should be considered as a PA. This sentence is debatable. Hence some explanatory statement must beaded.

b. If the aim of the study is to study the association between yoga practice and MVPA, then a correlation study will be enough to establish the aim. On the other hand, several factors are discussed that primarily affects the MVPA not yoga as such. The aim need to be broadened. Thus the title must also be changed.


Measure:
a. What was the method employed to interpret the PA data? (You can take example from IPAQ - short form data analysis format)

b. The style of use of any of the yoga technique decides the energy cost of the yoga. Therefore, “The compendium of physical activities” did not specify the energy costs of each of yoga(s). Please give reference of classification of yoga fitting into the classification you have used i.e. light activity , moderate activity & Vigorous activity.

Statistical analysis:

a. Why unequal class intervals are taken in age cohorts (5-10 years, 11-13 years & 14-17) ?

Result:
a. At many places SD is more than 2-3 times as per table 2: Please explain

b. I propose one table for univariate analysis to show the strength of relationship between the exposure and outcome variables and one multivariable analysis discussing the results of final model based on regression so that the predictors of yoga practice can be clearly elaborated. Otherwise, in the present form a question arises “on what basis the factors are chosen for the final model” . We recommend that the tables should contain the P value and beta values separately so that it’s easy on eye to capture the relationship among the data.

Discussion:

a. Discussion should be the explanation of your results associating with other studies on the same issue. In discussion section for example: Lane 186-192 discuss what other studies on the same have pointed out that supports or different from your findings and why you think the difference lies as per your research. Lane 187-192 can easily be put as policy suggestion.

b. Lane 200-201 By examining the effects of yoga practice on children and youth’s MVPA levels, this study contributes to addressing this gap in knowledge.

c. In your study how Yoga practice affects the MVPA levels is not addressed properly. An odds ratio with a corelation analysis would have been better options.

d. Can you explain why there is no significant association between ‘Average minutes of yoga practice per day’ and age group 11-13 year. Similarly rural and urban younger populations in yoga practice?

e. How do you explain the non-significant association of with MVPA with children of 14-17 years having active friends? Similarly what would be your explanation of non-significant association of with MVPA with rural dwelling children despite having active friends?

f. In the later part of the discussion for example lane 214-221, the relation of MVPA with that of “having active friends” is focused. There is no discussion on relation between yoga & “having active friends”.

g. Yoga has a very ancient spiritual root in India. Hermits practiced them in remote areas and stayed away from the habitations. Then why yoga practice has a better foothold in urban areas than rural area? What are your thoughts please share.

h. Rural people and rural communities are said to be the flag bearer of cultural traditions of India. This study points out that rural children have lesser chances of adapting yoga practice. Still then why the authors recommend “culturally appropriate” PA promotion referring to yoga practice. In addition, yoga practice is just another form of PA (as listed in the questionnaire prepared by the authors), what led the authors conclude the lane 266-267.

i. What is the proportion of children who practiced breathing exercise and those who practiced ashtanga or vinyasa yoga. How many practiced both. Ashtanga yoga encompass both pranayama & postures with meditation. What is the proportion of children practiced Yogic PA only who met MVPA and non-yogic PA who met MVPA and those who practiced both yogic and non-yogic variety of PA who met MVPA?


About conclusion:

a. Already the Yoga in inducted into the PA guideline of India i.e. “Fit India Moment” across all age groups. Hence your study only re-enforces a decision which has already been taken by the policy makers of PA at pan-India level.

Experimental design

A part of a ‘muti centre cohort’ study is confusing. Is the original study in any form being published ? Please detail the original ‘muti centre cohort’ study in 250 words explaining how this cross sectional observational survey study is fitting into the ‘muti centre cohort’ study.

However, Observational cross sectional design is employed. No issues hare.

Sample size calculation:
a. Sample size calculation methodology is not discussed at all.

Data collection:
a. Please elaborate the stage how the schools were chosen? Why there are so less schools from the rural areas in comparison?

b. To our experience, the CBSE, ICSE pattern and other local board patterns differ in curriculum with very different stress on physical activity promotion patterns. How did you ensure a homogeneity in samples drawn?

On the questionnaire:

a. Was your instrument (questionnaire) a pre-tested questionnaire? Did you use a standardised instrument (questionnaire) to extract PA data ? If No, then did you test the validity or reliability of your PA instrument (questionnaire) ?

b. We feel two important questions are missing from the questionnaire: 1. One question on whether yoga practice is part of school PA program or not 2. another question regarding inter-school yoga competition would have been quite informative.

c. It is highly unlikely to get a very high response pattern from schools (41 out of 50 schools that is almost 82%). Did you employ any method to pursue school principals?

d. Many of the questionnaire items are not discussed in the results. Hence, those questions not relevant to this article is confusing.

e. Please define MVPA with specific reference to the questionnaire you have used.

Validity of the findings

a. At many places SD is more than 2-3 times as per table 2: Please explain
b. Please define MVPA with specific reference to the questionnaire you have used.

Additional comments

No

·

Basic reporting

NA

Experimental design

More clarification required on sample selection.

Validity of the findings

more reports required on rural , urban, family income, public/ private schools, number of siblings in family

Additional comments

The authors have cateogrised the children into two groups- having active friends and not having active friends. However, as many children would know they do not have any active friend, this maybe a demotivator for all. There is a need to provide healthcare education to those who are not active friends.
A copy of questionnaire that the authors have used has not been shared. A copy of the same needs to be shared. Whether it is a standard tool or modified tool for a new tool has to be validated.
There is a difference between physical activity level in the rural and urban areas. As in the rural areas, children are more involved in outdoor games and sports, it seems that the authors have not considered these aspects.
While the authors mentions about the different forms of Yoga- physical postures (asanas), breathing techniques 60 (pranayama), and meditation practices, it is important to understand that not all Yoga are related to Physical activity. Pranayama and Meditation practices does not translate to physical activity. However, they maybe a good indicator for mental wellbeing and respiratory wellbeing. Have the authors only looked for physical posters or a mix of the three , that has to be specified.
The study also does not inform whether the schools were located in urban or rural areas, there is also a gap with the information on whether the schools were government or private schools.
This is a very important study and the authors may like to improve the colloquial phrases from the above study. Still more outputs could be gained from this study.

---

## Round 0.2 · Minor Revisions

Dear Dr. Patel and colleagues:

Thanks for revising your manuscript. The reviewers are very satisfied with your revision (as am I). Great! However, there are a few issues to entertain. Please address these ASAP so we may move towards acceptance of your work.

Best,

-joe

Reviewer 2 ·

Basic reporting

Improved yet some more clarification required

Experimental design

Sampling is yet a doubtful area, i have raised few questions again.

Validity of the findings

Author's claims are genuine but yet few questions remain.

Additional comments

Respected sir, I am Reviewer 2 : I am here with attaching my follow up comments which are mentioned under the rebuttal of the authors. I have some additional comments for the authors.

Reviewer 2

1. Lane 33-34: Please explain this “However, there is little evidence of the relationship between yoga practice and physical activity”. According to us if we consider the definition of physical activity (PA) yoga should be considered as a PA. This sentence is debatable. Hence some explanatory statement must be added.
Thank you for your feedback. We agree that yoga, like active transportation, and participation in sports, is a type of physical activity, which can contribute to the accumulation of MVPA – a key indicator of health according to WHO guidelines. The focus of this study was to understand if there is a relationship between yoga practice and MVPA so that we can inform culturally appropriate physical activity programming that would increase MVPA levels.

We have now added from lines 226 to 232: “Although no studies have examined the association between yoga practice and MVPA among children and youth in India, these findings are consistent with previous research that has demonstrated the positive effects of yoga interventions on various aspects of health among children and youth in India, including a study which found that yoga interventions can improve physical, cognitive, and emotional measures among children in India. In addition, a recent study by Khadilkar et al (2023) found that yoga is comparable to physical exercise in influencing muscle function, highlighting the broader impact of yoga on health that goes beyond MVPA.”.

Reviewer: Accepted.

2. If the aim of the study is to study the association between yoga practice and MVPA, then a correlation study will be enough to establish the aim. On the other hand, several factors are discussed that primarily affects the MVPA not yoga as such. The aim need to be broadened. Thus the title must also be changed.
Thank you for this suggestion. The primary independent variable of interest here, which addresses a key gap in evidence, is yoga practice. Standard regression analysis practices were applied to control for socio-demographic and other confounders (i.e., related contextual factors) that moderate the relationship between yoga and MVPA. We have now added to the Strengths and Limitations (lines 343 to 347): “Additionally, controlling for motivating and facilitating factors for physical activity is critical to ensure that the association between yoga practice and MVPA is valid. Future studies should substantiate our study findings with longitudinal studies and randomized controlled trials to control for additional unobserved confounders.”.

As suggested, we have now broadened the aim by adding: “This study aims to investigate the association between yoga practice and MVPA among children and youth in India, while controlling for sociodemographic and related contextual factors, to inform culturally appropriate PA interventions.” (lines 67 to 69). We have also now broadened the title to capture this context: “Association between yoga and related contextual factors with physical activity among children and youth: a culturally appropriate path to reduce the burden of non-communicable diseases in India” (lines 1 to 3).

Reviewer: Accepted.

3. The style of use of any of the yoga technique decides the energy cost of the yoga. Therefore, “The compendium of physical activities” did not specify the energy costs of each of yoga(s). Please give reference of classification of yoga fitting into the classification you have used i.e. light activity , moderate activity & Vigorous activity.

Thank you for your feedback. The objective of our study was to understand if any form of yoga was associated with physical activity among children and youth in India. As there is no existing evidence that associates yoga practice with MVPA in Indian children and youth, the focus of this study was to see if there is that relationship, irrespective of the type/intensity of yoga practice. While we acknowledge that there are varying levels of intensity to classify yoga, we did not make this distinction in our study. However, we have now added in the Strengths and Limitations section: “Although previous research indicates there are varying levels of intensity to classify yoga (i.e., asanas and pranayama), these variations were beyond the scope of our study. Future studies should compare these associations between yoga practice and MVPA across varying intensities of yoga practice (e.g., light activity for breathing yoga or moderate-to-vigorous activity for physical yoga) to capture nuanced variations in the relationship between yoga and MVPA.” (lines 347 to 351).
Reviewer: Accepted.

4. Why unequal class intervals are taken in age cohorts (5-10 years, 11-13 years & 14-17)?
Thank you for raising this question. We have now added a statement in the Measures which outlines our rationale for choosing the three age cohorts: “We further categorized individuals into three age groups (aged 5 to 10 years, aged 11 to 13 years, and aged 14 to 17 years), representing the different developmental stages from childhood to adolescence. In particular, the World Health Organization defines individuals under 10 years of age as children, whereas individuals over 10 years of age are considered adolescents[28]. Llorente-Cantarero et al (2020) found that physical activity and sedentary behaviour increased in the transition from childhood to adolescence [29]. Moreover, there are distinct physical, sexual, cognitive, social, and emotional changes that occur between early adolescence (ages 10 to 13) and middle adolescence (ages 14 to 17) that may influence physical activity [30].” (lines 105 to 113).


Reviewer 2: Ref 30 is ref 29 in the supplied reviewed manuscript. (Further Check needed)

5. At many places SD is more than 2-3 times as per table 2: Please explain
Thank you for your comment. The high standard deviations may be attributable to the wide range of values for physical activity and yoga practice because active living behaviours may vary widely across youth. Moreover, this is supported by previous evidence which also found that child and youth MVPA has high variability, even when measured using objective tools such as accelerometers [1].

Reviewer 2: How do you ensure homogeneity in the sample? Is the statistical test employed in this study is robust to accommodate this kind of variability? Is the basic assumption in employing this statistical test (Logistic Regression analysis in your case) are adhered to?

6. I propose one table for univariate analysis to show the strength of relationship between the exposure and outcome variables and one multivariable analysis discussing the results of final model based on regression so that the predictors of yoga practice can be clearly elaborated. Otherwise, in the present form a question arises “on what basis the factors are chosen for the final model”. We recommend that the tables should contain the P value and beta values separately so that it’s easy on eye to capture the relationship among the data.

Thank you for your feedback. We recognize the importance of using a rigorous approach
to select the factors included in a model. We have now added a table to the Supplementary Files
which includes univariate analyses capturing the strength of the relationship between each
of the exposure variables with MVPA (the outcome variable). We have also provided the P-values
and beta values separately as suggested. Controlling for motivating/facilitating factors of physical
activity in the multivariate analyses was critical to ensure that the association between yoga
practice and MVPA is valid. The data collection strategy was intended to capture as many
behavioural, contextual (social, physical) and socioeconomic factors as possible that influence PA – this strategy was informed by key indicators of the 2016, 2018, and 2022 report cards. The final model included variables that were deemed theoretically important and aligned with our
research questions.

Reviewer: Accepted. For entry into the multi variable analysis the univariable analysis should show statistically significant association or at least the P value should be around 0.08. (This is a standard practice).

7. Discussion should be the explanation of your results associating with other studies on the same issue.
In discussion section for example: Lane 186-192 discuss what other studies on the same have pointed out that supports or different from your findings and why you think the difference lies as per your research.
Lane 187-192 can easily be put as policy suggestion.
Lane 200-201 By examining the effects of yoga practice on children and youth’s MVPA levels, this study contributes to addressing this gap in knowledge.
Thank you for your feedback. We have now improved the Discussion by including previous literature that substantiates our main study findings: “This study found that yoga practice was associated with higher MVPA levels among children and youth in diverse locations across India. Our study findings align with the 2022 India Report Card, which emphasizes promoting physical activity through yoga programs for children and youth. The report not only supports our main conclusion but also recommends strategies for increased engagement in yoga, highlighting the holistic benefits of integrating yoga into physical activity initiatives [13]. Although no studies have examined the association between yoga practice and MVPA among children and youth in India, these findings are consistent with previous research that has demonstrated the positive effects of yoga interventions on various aspects of health among children and youth in India, including a study which found that yoga interventions can improve physical, cognitive, and emotional measures among children in India [14]. In addition, a recent study by Khadilkar et al (2023) found that yoga is comparable to physical exercise in influencing muscle function, highlighting the broader impact of yoga on health that goes beyond MVPA [45].” (lines 22 to 232).

Reviewer: Accepted. But discussion as per the clean document starts from line 206

8. In your study how Yoga practice affects the MVPA levels is not addressed properly. An odds ratio with a correlation analysis would have been better options.
Thank you for your feedback. We considered examining how yoga practice is associated with the odds of meeting the MVPA guidelines using logistic regression. However, we believe that using a continuous outcome variable (MVPA) enabled us to gain a more nuanced understanding of the relationship between yoga practice and MVPA. For instance, our study found that the daily minutes of yoga practice was associated with higher daily minutes of MVPA, which highlights that promoting yoga practice can be a promising strategy to increase MVPA.

Moreover, our standard regression analysis practices were applied to control for socio-demographic and other confounders (i.e., related contextual factors) that moderate the relationship between yoga and MVPA, which would not have been possible with simple correlation analyses.

Reviewer: Accepted.

9. Can you explain why there is no significant association between ‘Average minutes of yoga practice per day’ and age group 11-13 year. Similarly rural and urban younger populations in yoga practice?
Thank you for your question. We have now added a paragraph citing previous evidence that provides a potential explanation for why there is no significant association between yoga practice and MVPA in the 11-13 age group: “Moreover, this discrepancy may be attributable to the 11 to 13 age range representing a transition from childhood to adolescence, which involves significant physical, social and cognitive changes. For instance, a study by Cozett et al (2016) analyzed factors influencing physical activity among 11 to 13-year-olds and found that multiple factors are associated with changes in physical activity in this age group, such as parental influence, peer influence, perceived physical activity self-efficacy and perceived physical activity competence” (lines 238 to 243).

Additionally, in lines 298 to 306, we not only hypothesized why yoga was associated with MVPA in urban but not rural residents, but we also provided suggestions to address these disparities: “In urban areas, where organized sports and structured exercise programs are more accessible than in most rural areas, individuals practicing yoga are more likely to engage in other forms of physical activity [58]. Promoting physical activity tailored to rural communities' unique needs and circumstances may be necessary to address these disparities. For instance, in rural areas, children and youth may be more involved in outdoor games and sports than their urban counterparts [59]. Community-based interventions that focus on creating safe spaces for physical activity in rural areas, such as playgrounds and sports fields could help to increase physical activity levels among children and youth living in rural areas.”.

10. How do you explain the non-significant association of with MVPA with children of 14-17 years having active friends? Similarly what would be your explanation of non-significant association of with MVPA with rural dwelling children despite having active friends?
Thank you for your question. We have now added a potential explanation for the association between active friends and MVPA not being significant in the 14-17-year age group (lines 255 to 259): “This may be attributable to younger children engaging in more group-based physical activity, whereas older adolescents may have a higher degree of autonomy and independence in their activity choices, which may reduce the impact of peer influence. For instance, Smith et al (2022) found that organized and team-based physical activity was more common among younger children, and the participation rates declined with chronological age.”

Reviewer: A good explanation. Accepted.


11. In the later part of the discussion for example lane 214-221, the relation of MVPA with that of “having active friends” is focused. There is no discussion on relation between yoga & “having active friends”.
Thank you for your comment. Given that the primary outcome variable for this study was MVPA, assessing how having active friends is associated with yoga practice was beyond the scope of our study. We have now added an excerpt that more clearly defines our study objectives to clarify the scope of our study: “This study aims to investigate the association between yoga practice and MVPA among children and youth in India among children and youth in India, while controlling for sociodemographic and related contextual factors, to inform culturally-appropriate PA interventions.” (lines 67 to 69).

Reviewer: We accept your response. However, we feel this is an essential part of the discussion. We suggest to add a line or two explaining the association of “yoga & active friends” then come to discuss “yoga & MVPA”.

12. Yoga has a very ancient spiritual root in India. Hermits practiced them in remote areas and stayed away from the habitations. Then why yoga practice has a better foothold in urban areas than rural area? What are your thoughts please share.
Thank you for your comment. We acknowledge that yoga practice has ancient spiritual roots in India. We have added a section in the Discussion to discuss why urban regions may have higher levels of yoga than rural regions despite yoga being originally practiced in remote regions. (lines 291 to 297) “These findings align with a previous study conducted by Mishra et al (2020) which found that there was a higher prevalence of yoga practitioners in urban areas compared to rural areas of India [55]. Given that yoga has ancient spiritual roots in India, where it was originally practiced in remote regions [56], these findings highlight the contemporary shift in yoga’s prevalence. This shift in yoga prevalence may be attributable to yoga being viewed as a lifestyle activity for physical and mental well-being, rather than solely limited to the spiritual and meditative importance [57].

Reviewer: We accept your response.

13. Rural people and rural communities are said to be the flag bearer of cultural traditions of India. This study points out that rural children have lesser chances of adapting yoga practice. Still then why the authors recommend “culturally appropriate” PA promotion referring to yoga practice. In addition, yoga practice is just another form of PA (as listed in the questionnaire prepared by the authors), what led the authors conclude the lane 266-267.
We acknowledge and agree that rural communities may play a significant role in the cultural traditions of India. We have now provided a potential explanation on why children and youth living in urban regions may engage in more yoga practice: “Given that yoga has ancient spiritual roots in India, where it was originally practiced in remote regions, these findings highlight the contemporary shift in yoga’s prevalence. This shift in yoga prevalence may be attributable to yoga being viewed as a lifestyle activity for physical and mental well-being, rather than solely limited to the spiritual and meditative importance.” (lines 293 to 297).

Additionally, the cultural significance of yoga in urban regions may manifest in different ways, such as the incorporation of traditional elements in modern practices. The cultural appropriateness of yoga is also highlighted in a review by Bussing et al (as cited in our study), which suggested that clinical trials involving yoga as an intervention may not have received the same compliance with patient populations outside of India [2]. Moreover, we acknowledge that yoga practice can be conceptualized as a form of physical activity. However, this study is more focused on how the practice of yoga influences the participation of children and youth in other forms of physical activity. To make the gap clearer in the Abstract, we have rewritten and improved the structure of the sentence by adding (lines 26 to 28): “However, there is little evidence on whether yoga practice is associated with other forms of physical activity.”.

Reviewer: We accept your response.

14. What is the proportion of children who practiced breathing exercise and those who practiced ashtanga or vinyasa yoga. How many practiced both. Ashtanga yoga encompass both pranayama & postures with meditation. What is the proportion of children practiced Yogic PA only who met MVPA and non-yogic PA who met MVPA and those who practiced both yogic and non-yogic variety of PA who met MVPA?
Thank you for your feedback. Given that there is a dearth of evidence in this research area, the objective of our study was to understand if any form of yoga was associated with physical activity among children and youth in India. While we acknowledge that there are varying levels of intensity to classify yoga, we did not make this distinction in our study. However, we have now added: “Although previous research indicates there are varying levels of intensity to classify yoga (i.e., asanas and pranayama), these variations were beyond the scope of our study. Future studies should compare these associations between yoga and MVPA across varying intensities of yoga (e.g., light activity for breathing yoga or moderate-to-vigorous activity for physical yoga) to capture nuanced variations in the relationship between yoga and MVPA.” (lines 347 to 351).

Reviewer: We accept your response.

15. Already the Yoga in inducted into the PA guideline of India i.e. “Fit India Moment” across all age groups. Hence your study only re-enforces a decision which has already been taken by the policy makers of PA at pan-India level.
Thank you for your comment. The recommendations outlined in our study go beyond
encouraging policymakers to induct PA guidelines. The importance of our study is that
we highlight nuanced variations in yoga practice across age, gender, and location, which
have not previously been researched. For instance, in the Conclusion we mentioned:
“The results support the need to develop age-, gender-, and location-specific strategies, as
well as social support and school infrastructure strategies to promote MVPA.
Interventions incorporating sociocultural factors into community programs and school
curricula, such as yoga programs, must be developed. Overall, addressing physical
inactivity and promoting active living among children and youth in India requires a
multifactorial approach which is crucial for reducing the burden of NCDs
and improving the health and well-being of future generations.” (lines 371 to 376).

Reviewer: We accept your response, but overall, I don’t see any new direction. Please suggest the changes or additions to what is already there in “Fit India moment”. This way this paper will attract more policy makers to amend newer insights. We suggest the authors to have a look of the “Fit India moment” scheme of things and be clear in their policy suggestions.

Further, if the authors chose to stick to their rebuttal, they may add that their findings are reenforcing the “Fit India moment” scheme.

16. A part of a ‘muti centre cohort’ study is confusing. Is the original study in any form being published ? Please detail the original ‘muti centre cohort’ study in 250 words explaining how this cross sectional observational survey study is fitting into the ‘muti centre cohort’ study. However, Observational cross sectional design is employed. No issues hare.
This current cross-sectional, observational study is an extension of the original multicentre study. The multicentre school-based study commenced in 2016 and utilized a multi-stage stratified random sampling method to collect data from children and youth aged 3 to 18 from across six Indian states from 40 selected schools. A total of 14,339 children and youth were selected after excluding those with diabetes, serious illness, chronic disorders, and height or weight under 3rd and above the 97th percentile.
The original multicentre study encompassed a wide range of health-related parameters, including physical characteristics, nutritional status, and lifestyle factors – we have now cited additional studies to further clarify this point. This current study provides a snapshot of yoga practice and physical activity levels during the Coronavirus disease pandemic lockdown in 2021 from 1042 children and youth. By leveraging the data from the original multi-centre study, this study offers a unique perspective on physical activity patterns during a time of restrictions and limited outdoor opportunities.

Reviewer’s comment: Now as per the author’s explanation, it seems that the sampling universe was 14,339 out of which 1042 samples ware drawn. Please provide further information how the samples were drawn?. This may be linked to the wide variations noted (as per in point no 5).

17. Sample size calculation methodology is not discussed at all.
Thank you for bringing this to our attention, we included a sample size calculation methodology in the methods section by adding the following “A sample size calculation was carried out to achieve a 95% confidence level and a 5% margin of error, which resulted in a sample of 385 participants. Nevertheless, this study engaged with a total of 1042 children.” (lines 94 to 96).

Reviewer’s comment: Changes accepted.

18. Please elaborate the stage how the schools were chosen? Why there are so less schools from the rural areas in comparison?
Thank you for your questions. The lower proportion of schools in the rural areas (~40%) vs urban areas (~60%) may have been due to factors beyond our control, including logistical challenges, non-response in rural regions, and disparities in infrastructure. However, we have now added more details in the Recruitment and Participants section of the Methods to elaborate on the stages for school selection: “The multi-staged stratified random sampling procedure involved randomly selecting five out of the 28 Indian states (Gujarat, Madhya Pradesh, Maharashtra, Tamil Nadu, and Telangana). Then, a city (urban area) and a neighbouring village (rural area) were randomly selected from each state. A list of 50 schools was generated, with village schools being government public schools. School principals were approached with study information and invited to participate in the study. Of the 50 schools invited, 41 schools participated, with the primary reason for refusal being that the schools did not have time to participate in the study.” (lines 81 to 87).

Reviewer’s comment: Changes accepted.

19. To our experience, the CBSE, ICSE pattern and other local board patterns differ in curriculum with very different stress on physical activity promotion patterns. How did you ensure a homogeneity in samples drawn?
We did not specifically sample or collect data on the variations across school boards. However,
we recognize the importance of controlling for variations in physical activity promotion
patterns across school boards and have included this limitation in the Strengths and Limitations:
“This study did not distinguish between public and private educational institutions, which may be
important for understanding the factors that are related to MVPA levels among youth. In particular,
previous literature has found that in India, low socioeconomic status has been associated with a
decrease in physical activity [71]. Indian youth with higher family income were more likely to
private school institutions, where students have been found to engage in less MVPA than their
public school student counterparts [72]. Additionally, having an understanding of whether yoga
practice accumulated was part of a school physical activity program and whether their school
participated in inter-school yoga competitions, which could potentially be influenced by the type
of school children and youth attend, would have provided a more nuanced understanding of the
relationship between yoga practice and MVPA. Future studies should examine how these
associations between yoga practice and MVPA differ across various local school boards which may
have varying physical activity promotion patterns.” (lines 353 to 364).

Reviewer’s comment: Changes accepted.

20. Was your instrument (questionnaire) a pre-tested questionnaire? Did you use a standardised instrument (questionnaire) to extract PA data ? If No, then did you test the validity or reliability of your PA instrument (questionnaire) ?; Please define MVPA with specific reference to the questionnaire you have used; What was the method employed to interpret the PA data? (You can take example from IPAQ - short form data analysis format)
Thank you for this question. We have now added a statement and cited the validated questionnaire in the Measures: “Physical activity was assessed using a modified version of the International Physical Activity Questionnaire – Short Form [31], which asked participants about the weekly frequency and duration in minutes per day spent on various activities, including sports, weight training, running, and other disparate activities, such as jumping ropes, playing garden games, and swimming. The physical activity data were categorized into three intensity types: inactivity, light activity, and MVPA. Studies have found the Intraclass Correlation Coefficient for self-reported MVPA among children and youth to range from fair to good (range 0.565–0.78) [32,22]. MVPA was calculated as the combined time spent in moderate and vigorous activities.” (lines 118 to 126).

Reviewer’s comment: Changes accepted. In table S1 no form of YOGA is mentioned under PA vigorous. We acknowledge the fact that this is a standard IPAQ. How do you suggest to mitigate this?


21. We feel two important questions are missing from the questionnaire: 1. One question on whether yoga practice is part of school PA program or not 2. another question regarding inter-school yoga competition would have been quite informative.
Thank you for your questions. We acknowledge that having an understanding of whether yoga practice accumulated was part of a school PA program and whether their school participated in inter-school yoga competitions would have provided a more nuanced understanding of the relationship between yoga practice and MVPA. We have now added a statement in the strengths and limitations section: “Additionally, having an understanding of whether yoga practice accumulated was part of a school PA program and whether their school participated in inter-school yoga competitions, which could potentially be influenced by the type of school children and youth attend, would have provided a more nuanced understanding of the relationship between yoga practice and MVPA. Future studies should examine how these associations between yoga practice and MVPA differ across various local school boards which may have varying physical activity promotion patterns.” (lines 358 to 362).

Reviewer’s comment: In the absence of those crucial components from the body of the questionnaire is likely to introduce a bias.

22. It is highly unlikely to get a very high response pattern from schools (41 out of 50 schools that is almost 82%). Did you employ any method to pursue school principals?
Thank you for your question. We were able to achieve a high response rate due to the established relationships we built with schools and educational institutions prior to the study, including previous collaborations and involvement in our multi-centre study in previous years.

Reviewer’s comment: Changes accepted.

23. Many of the questionnaire items are not discussed in the results. Hence, those questions not relevant to this article is confusing.
Thank you for your comment. We have described each of the questionnaire items introduced in the measures section in the results. We also provided our full questionnaire in the appendix to provide a more detailed understanding of the exact questions. We have now bolded the questions we used in our study to make the relevant variables for this specific study more clear.

Reviewer’s comment: Changes accepted.

Additional comments:
1. Title still requires modifications. For example - Association between yoga and related contextual factors with moderate to vigorous physical activity among children and youth aged 5-17 years in five Indian states.

We suggest to consider deleting “A culturally appropriate path to reduce burden of non-communicable disease in India”.

2. Comment on Table S2. Incomplete table presented. For example, the Age categories are missing along with the proportions, Odds ratio with CI is missing. There is a lot of scope of improvement in table S2.

3. Lane 24-25 “Physical activity's role in reducing non-communicable disease risk among children and youth is well established”. Please be specific which NCD risk factors are present in children & youth except physical inactivity.

4. The results of unpaired t-test & Regression is discussed. If I am not wrong, we are missing the results of ANOVA & the post-hoc analysis.

·

Basic reporting

No issues

Experimental design

No issues

Validity of the findings

No reported concerns

Additional comments

NA

---

## Round 0.3 · accepted · Accept

Dear Dr. Patel and colleagues:

Thanks for revising your manuscript based on the concerns raised by the reviewer. I now believe that your manuscript is suitable for publication. Congratulations! I look forward to seeing this work in print, and I anticipate it being an important resource for groups studying the health benefits and limitations of Yoga. Thanks again for choosing PeerJ to publish such important work.

Best,

-joe